# Genomic Basis for Individual Differences in Susceptibility to the Neurotoxic Effects of Diesel Exhaust

**DOI:** 10.3390/ijms232012461

**Published:** 2022-10-18

**Authors:** Alexandra Noël, David G. Ashbrook, Fuyi Xu, Stephania A. Cormier, Lu Lu, James P. O’Callaghan, Shyam K. Menon, Wenyuan Zhao, Arthur L. Penn, Byron C. Jones

**Affiliations:** 1Department of Comparative Biomedical Sciences, School of Veterinary Medicine, Louisiana State University, Baton Rouge, LA 70803, USA; 2Department of Genetics, Genomics, and Informatics, College of Medicine, University of Tennessee Health Science Center, Memphis, TN 38163, USA; 3School of Pharmacy, Binzhou Medical University, Yantai 264003, China; 4Department of Biological Sciences, Pennington Biomedical Research Center, Louisiana State University, Baton Rouge, LA 70808, USA; 5Molecular Neurotoxicology Laboratory, Toxicology, and Molecular Biology Branch, Health Effects Laboratory Division, Centers for Disease Control and Prevention, NIOSH, Morgantown, WV 26508, USA; 6Department of Mechanical and Industrial Engineering, School of Veterinary Medicine, Louisiana State University, Baton Rouge, LA 70803, USA

**Keywords:** diesel exhaust, neurotoxicity, neuroinflammation, strain, sex, pollution, environmentally persistent free radicals

## Abstract

Air pollution is a known environmental health hazard. A major source of air pollution includes diesel exhaust (DE). Initially, research on DE focused on respiratory morbidities; however, more recently, exposures to DE have been associated with neurological developmental disorders and neurodegeneration. In this study, we investigated the effects of sub-chronic inhalation exposure to DE on neuroinflammatory markers in two inbred mouse strains and both sexes, including whole transcriptome examination of the medial prefrontal cortex. We exposed aged male and female C57BL/6J (B6) and DBA/2J (D2) mice to DE, which was cooled and diluted with HEPA-filtered compressed air for 2 h per day, 5 days a week, for 4 weeks. Control animals were exposed to HEPA-filtered air on the same schedule as DE-exposed animals. The prefrontal cortex was harvested and analyzed for proinflammatory cytokine gene expression (*Il1β*, *Il6*, *Tnfα*) and transcriptome-wide response by RNA-seq. We observed differential cytokine gene expression between strains and sexes in the DE-exposed vs. control-exposed groups for *Il1β*, *Tnfα*, and *Il6*. For RNA-seq, we identified 150 differentially expressed genes between air and DE treatment related to natural killer cell-mediated cytotoxicity per Kyoto Encyclopedia of Genes and Genomes pathways. Overall, our data show differential strain-related effects of DE on neuroinflammation and neurotoxicity and demonstrate that B6 are more susceptible than D2 to gene expression changes due to DE exposures than D2. These results are important because B6 mice are often used as the default mouse model for DE studies and strain-related effects of DE neurotoxicity warrant expanded studies.

## 1. Introduction

### 1.1. Air Pollution Produces Multi-Organ System Disease

Air pollution has been known to be an environmental health hazard for many decades [1,2]. Airborne pollutants are mainly composed of carbonaceous particles to which a mixture of inorganic and organic compounds is adsorbed. Thus, particulate matter (PM) consists of coarse (<10 µm; PM_10_), fine (0.1–2.5 µm; PM_2.5_) and ultrafine (< 0.1 µm; PM_0.1_) carbonaceous particles [3]. The United States Environmental Protection Agency (US EPA) and the World Health Organization (WHO) have established annual average standards for PM_2.5_ exposures, which are set at 12 and 5 µg/m^3^, respectively [4]. A major source of air pollutants is transportation and especially diesel-powered vehicles. Diesel exhaust (DE) is a complex mixture of gasses and PM. These particles consist of aromatic aryl hydrocarbons and other matter [5,6]. In addition, the negatively charged particles are free to bind to and carry metal oxides and ions, especially manganese, iron, copper, and zinc [7]. Furthermore, the metal oxides allow for the chemisorption of organics to particulates in DE forming a pollutant particle containing a stabilized surface free radical known as environmentally persistent free radicals (EPFRs) [8]. EPFRs undergo Fenton chemistry, cyclically producing toxic hydroxyl radicals when taken up by cells and have documented lifetimes of at least 5 years, depending on the source [7].

### 1.2. Particulate Components of Air Pollution and Diesel Exhaust Are Particularly Troublesome

The concentration of individual air pollutants as well as the size of particles and agglomerates vary according to environmental conditions and affect biological responses [9]. In the United States, in 2020, the mean concentration of PM_2.5_ was 8.02 µg/m^3^ (US EPA, 2021) [10]. According to the American Lung Association [11], nearly 135 million Americans are living in areas where air pollution, mainly characterized by ozone and PM_2.5_, can reach critical levels (e.g., 55.5 to 150.4 µg/m^3^ for PM_2.5_). In the United States, DE particle concentrations have been shown to range between 20 and 25 μg/m^3^ along transportation routes [5,12]. In addition, under experimental conditions, it has been reported that maximum concentrations of 350 μg/m^3^ of DE particles can be generated in the vicinity of a diesel engine vehicle [13]. These data highlight that individuals can be acutely exposed to unhealthy levels of PM_2.5_, including DE particles. In fact, in 2021, it was estimated that outdoor air pollution accounted for nearly 4.2 million premature deaths worldwide [14]. Epidemiological data have shown that air pollution is associated with decreased lung function in healthy and susceptible populations of all ages [15,16,17,18,19,20]. It is also a problem for asthmatics and others who may be allergic to its constituents [21]. It is now well established that the ultrafine component of air pollution causes adverse pulmonary effects through mechanisms associated with inflammation and oxidative stress [22,23,24,25,26,27]. Mechanistic studies have shown that inhaled fine and ultrafine particles induce inflammation in the lungs [28]; and it is thought that the pro-inflammatory mediators, locally produced, subsequently enter the bloodstream and contribute to systemic inflammation [29,30,31,32,33,34]_._ It was further shown that insoluble inhaled ultrafine particles deposit into the lower respiratory tract and from there, undergo translocation to the blood and extra-pulmonary organs [35,36]. However, a defined set of factors, such as primary particle size and structure, surface charge, chemical composition, and the agglomeration state, have been shown to determine the capacity of PM_0.1-2.5_ to cross the air-blood barrier [37]. Thus, early research on air pollution, specifically on DE, has focused on respiratory morbidities; however, more recently, air pollution has been shown to affect, in addition to the lungs, multiple organ systems, including the central nervous system. 

### 1.3. Diesel Exhaust Exposure Carries Risk for Neurological Disease

Exposure to air pollution and especially DE is associated with neurological developmental disorders and neurodegeneration [38,39]. While the exact mechanisms are poorly defined, inflammation and oxidative stress are two of the mechanisms often cited in DE exposure toxicity, exactly how DE particles exert their influence on brain function is less clear. One proposed means is by the particles entering the brain through the olfactory bulbs [40] and another is through extra-pulmonary translocation as described above, where particles can enter the circulatory system and be delivered to the brain [41]. The functional effects of air pollutants on brain function include impaired cognition [42,43,44,45]. The pathophysiological features of DE neurotoxicity are increased proinflammatory cytokine gene expression, increased glial, activity, and indices of neurodegeneration. The impaired cognition in those exposed is likely caused by particulates reaching the brain [46]. Few preclinical papers relate this toxicity in terms of genetic or sex differences; however, there is some evidence for sex effects with male mice being more affected by DE neurotoxicity [47]. Although not addressed directly to neurobiological effects of air pollution, a recent report by Hϋls and colleagues [48] reported genetic variability in air pollution effects on the airway endoplasmic reticular stress system to confer differential risk for inflammatory response. One might expect similar gene-based differential susceptibility to inflammatory responses in other systems, including the central nervous system [47]. As noted by O’Callaghan and Miller [49], neurodegeneration is almost always accompanied by neuroinflammation; however, neuroinflammation can occur without neuropathology, as in the case of chronic sickness behavior.

### 1.4. Neurological Effects of Exposure to Diesel Exhaust: Not Everyone Is Equally Susceptible

A large proportion of the population living in urban areas is exposed to airborne pollutants, including DE particles. The latter represents a risk factor that is not of genetic origin or related to a modifiable lifestyle but is rather a factor that one cannot control or reduce the exposure individually. Accordingly, it is essential for policymakers to act promptly to reduce this public health burden. In this way, the necessity and originality of this study reside in the likelihood that not all individuals are equally susceptible to neurological damage caused by exposure to air pollution, especially DE [50]. Such information is of paramount importance for policymakers as these types of data must be considered in the establishment and periodical revision of air quality guideline values representing acceptable air pollution exposure levels for general and susceptible populations. Accordingly, we studied the effects of sub-chronic inhalation exposure to DE on neuroinflammatory markers in two inbred mouse strains and both sexes, we also included a whole genome examination of gene expression in the medial prefrontal cortex following DE exposure in the two strains and both sexes. Here, in this pilot study, we report strain- and sex-dependent differential neurological response to exposure to DE and present a mandate for broader genetic and genomic examination with translational implications.

## 2. Results

### 2.1. Proinflammatory Cytokine Gene Expression

Figure 1 shows gene expression for *Il1β*, *Il6*, and *Tnfα* by strain, sex, and DE exposure. For *Il1β*, we observed significant main effects for strain (F_1,22_ = 6.86, *p* < 0.02), DE (F_1,22_ = 4.43, *p* < 0.05), sex (F_1,22_ = 6.37, *p* < 0.02), and all interactions, viz., strain X DE (F_1,22_ = 6.83, *p* < 0.02), strain X sex (F_1,22_ = 6.41, *p* < 0.02) and strain X DE X sex (F_1,22_ = 6.88, *p* < 0.02). For *Il6*, we observed significant main effects for strain and sex (F_1,20_ = 18.03, *p* < 0.001; F_1,20_ = 11.96, *p* < 0.003, respectively) and strain X DE, strain X sex, DE X sex and strain X DE X sex interactions (F_1,20_ = 19.42, *p* < 0.001; F_1,20_ = 11.07, *p* < 0.004; F_1,20_ = 13.45, *p* < 0.003; F_1,20_ = 9.36, *p* < 0.007, respectively). The main effect of DE was not significant (F_1,20_ = 1.88, *p* < 0.20), having been obscured by the strain X treatment interaction. For *Tnfα*, we observed a significant main effect of strain (F_1,22_ = 5.83, *p* < 0.03), and interactions, strain X DE (F_1,22_ = 7.83, *p* < 0.02), strain X sex (F_1,22_ = 5.14, *p* < 0.04), and strain X sex X DE (F_1,22_ = 6.8, *p* < 0.02. *Tbp* expression did not change as a result of exposure to DE. Table 1 shows the gene expression results for the animals exposed to air.

### 2.2. Whole-Genome Gene Expression by RNA-seq

Having observed strain and treatment-specific effects on genes of interest, we next examined gene expression using RNA-seq. An *n* of 2 per strain, sex, and treatment combination was planned, however, principal components analysis revealed a strong outlier which we removed, resulting in the loss of a sample. Accordingly, a fully balanced analysis could not be carried out. To confirm the results of the qPCR, the same three genes were examined. Although all effects were not recapitulated, potentially due to lower power, we did observe a significant strain-by-DE effect (*p* < 0.05) on *Il1β*, a significant strain-by-sex effect on *Tnfα* by both measures (*p* < 0.05), and no significant effects on *Il6* (*p* > 0.05). Expression of *Gadph* was unaffected by DE exposure.

Figure 2 presents the number of differentially expressed genes, DE-air, by strain and sex. There are remarkable differences both for strain and sex. As mentioned above, the expression data were subjected to principal components analysis (PCA). The results are shown in Figure 3. The first principal component covered 32% of total variance and revealed one outlier that was removed from subsequent analyses.

We next examined each single and pairwise effect of strain, sex, and treatment on gene expression across the whole genome, and submitted all significantly differentially expressed genes to over-representation analysis using WebGestalt (Figure 4). We found that 213 genes were differentially expressed by DE main effect (Figure 4A), and these genes are highly enriched for annotations related to natural killer cell-mediated cytotoxicity (KEGG mmu04650 FDR < 0.033, GO:0042271 FDR < 0.00006). This is in agreement with previous findings [51,52]. In the D2 mice alone (Figure 4B), there were 150 genes differentially expressed between air and DE treatment, and again we see annotations related to natural killer cell-mediated cytotoxicity (KEGG mmu04650 FDR < 0.011, GO:0042271 FDR < 0.000025). In the B6 mice alone (Figure 4C), there were 218 genes differentially expressed between air and DE treatment, and again we see annotations related to natural killer cell-mediated cytotoxicity, although the KEGG pathway has dropped to sub-significance (KEGG mmu04650 FDR < 0.09, GO:0042271 FDR < 0.00014). In males (Figure 4D), there were 120 genes differentially expressed between air and DE treatment, but there are no significant annotations. In females (Figure 4E), there were 434 genes differentially expressed between air and DE treatment, and again, we see significant enrichment for annotations related to natural killer cell-mediated cytotoxicity (GO:0042271 FDR < 0.04), but also annotations related to ribosomes (KEGG mmu03010 FDR < 0.000002, Wikipathway WP163 FDR < 0.00004). In addition, there were only 9 genes differentially expressed between males and females when exposed to DE (Figure 4F), with no significantly enriched annotations. Furthermore, there were 648 genes differentially expressed between D2 and B6 mice exposed to DE (Figure 4G). We see annotations related to ribosomes (e.g., GO:0030490 FDR = 0.0001, WP163 FDR = 5.8833 × 10^−12^), and to cytoplasmic translation (GO:0002181 FDR = 0.00005). Lastly, 758 genes are differentially expressed between D2 animals and B6 animals in the control group (Figure 4H). These are also enriched for annotations related to ribosomes (GO:0030490 FDR = 0.0004, KEGG mmu03010 FDR = 6.4676 × 10^−10^, WP163 FDR = 3.4491 × 10^−9^). 

Finally, we performed an analysis to show the top 50 differentially expressed genes by strain and sex. Two genes, *Slc5a1* and *Fam19a3*, were equally responsive across both strains and sexes and 24 genes showed differential responses with B6 males and D2 females showing similar expression differentials while B6 females and D2 males were mostly similarly unresponsive. These effects are presented as a heatmap for absolute differential values in Figure 5.

## 3. Discussion

### 3.1. Support for the Hypothesis That Individual Differences in Susceptibility to the Adverse Health Effects of Exposure to Diesel Exhaust Are Related to Genetics

DE is a major source of air pollution that is known to cause adverse health effects, including in the respiratory tract and beyond, particularly in the central nervous system [1,2]. In this study, we investigated the effects of DE exposure on neuroinflammatory markers in aged mice of both strains and sexes. We found that sub-chronic exposures (2 h per day for 4 weeks) to DE resulted in strain and sex differential gene expression for the cytokine *Il1β* when compared to air-exposed control mice (Figure 1). Through RNA-sequencing of the medial prefrontal cortex, we found genome-wide differential expression and differential ontologies related to neurodegenerative diseases (Figure 4). Overall, our results show differential strain-related effects of DE on neurotoxicity and demonstrate that B6 mice are more susceptible, as evidenced by gene expression changes to DE (Figure 2). In addition, the protein-coding gene, engrailed homeobox 1 (*En1*), has been linked to lung cancer caused by benzo(a)pyrene in air pollution [53,54]. Furthermore, dead-box helicase 3Y-linked (*Ddx3y*), also a protein-coding gene, is upregulated by 9,10-phenanthraquinone, a component of DE particles [55,56,57], and is differentially expressed in those exposed to DE [58]. Overall, these results suggest that the increased susceptibility of B6 mice compared to D2 mice to DE exposures may be driven by the nitropyrene, hexanal, benzo(a)pyrene, and 9,10-phenanthraquinone, or EPFR, components of DE [59].

### 3.2. RNA-seq Identifies Genes and Pathways That May Confer Differential Susceptibility to the Neurotoxic Effects of Diesel Exhaust

There were 11 genes significantly differentially expressed in D2 between DE treated and air control, genes not significantly or suggestively differentially expressed in the B6. These include *Egfl8*, *Psmc3ip*, *Cmtm3*, *Npas4*, *Gm11843*, *Gm15903*, *Gm10538*, *Gm38031*, *Gm37632*, *Gm37963*, *and Gm29724*. To the best of our knowledge, none of these genes, including the protein-coding genes: EDF-like domain multiple 8 (*Egfl8*), Proteasome (prosome, macropain) 26S subunit ATPase3 interacting protein (*Psmc3ip*), and CKLF like marvel transmembrane domain containing 3 (*Cmtm3*), have been previously related to DE exposures [59]. Most importantly, the gene for neuronal PAS domain protein 4 (*Npas4*), which was uniquely dysregulated in the DE-treated D2 mice (Figure 4), has been identified as a transcription factor that plays significant roles in neural circuit formation and function, neuronal plasticity, in addition to having neuroprotective properties [58]. These data highlight the novelty and importance of our findings, where key genes related to neuroprotection against the development of neurological diseases, can be altered by DE exposures in a strain-specific manner. Generally speaking, our data implicate DE-related inflammatory responses in aging and suggest potential signaling targets that may facilitate neuroprotection through an influence on for example, hormesis [60]. Evidence of hermetic effects of toxins such as DE require careful dose response studies. While we may have altered expression of so-called hermetic vitagenes in this study, we cannot assert such evidence, having exposed our animals to only one dose. Nevertheless, the hermetic effect of DE very likely has genetically related individual differences and is an excellent candidate for future studies in genetic reference populations. In aggregate, our data show that there are genes known to be involved in responses to DE that are dysregulated in one strain (B6), but not in the other (D2). This supports the idea of strain-specific variations in response to DE and the need for more research to establish which mouse strains generate results that are most similar to human responses under DE exposure conditions. 

### 3.3. Overall Impact of This Study

The data presented here are the first of their kind and represent a forward genetics approach to the toxicogenetic study of airborne pollutants. The approach involves the rigorous definition of phenotypes followed by the search for genes that underlie the differential response of these genes to environmental perturbations, such as exposure to toxicants and toxins. The mice that were the subjects of this study come from two well-known inbred strains, viz., C57BL/6J and DBA/2J. These strains are the parentals of the family of BXD recombinant inbred strains in which the approximately 6 million genetic differences between the two are recombined in 140 derived inbred strains. It is common and efficient to conduct preliminary studies in the parental strains prior to employing multiple BXD strains, i.e., to show proof of principle prior to embarking on a large, expensive study. The aim of the study presented here was just that. As a result, we are now confident that there are sufficient biochemical differences between the parentals and importantly, between the sexes to warrant developing a more comprehensive study of 30–40 BXD strains. The aim of the expanded study is to nominate candidate genes underlying strain differences in the neurological effects of exposure to DE and to conduct genetic correlational analyses of the effects of DE among the strains against the more than 6000 phenotypes measured in the BXD panel that are listed at www.genenetwork.org (accessed on 5 May 2022). This is a publicly available database that contains phenotypes from hundreds of studies and also lists basal gene expression data for many tissues, including brain regions. 

### 3.4. Why Mice?

The European house mouse (*Mus musculus*) has served as human analogue in basic research for many decades. Ethical and logistic limitations preclude almost all toxicogenetic research in humans. Genome-wide association studies in humans have revealed the genetic basis for individual differences in several diseases; however, the exact mechanisms for gene action are difficult to ascertain. Thus, the use of animal models to uncover mechanisms becomes the approach [61,62]. Although there are important species differences between mice and humans, there is considerable overlap between the two genomes and what we learn in mice can inform the situation in humans [62] and thus become the basis for translational value of mouse research [63].

### 3.5. Summary

Overall, this study, elucidating strain differences between B6 and D2 mice in prefrontal cortex responses to DE exposures, lends proof of concept of genetic-based differential susceptibility to the neurotoxic effects of DE exposure. This is important, as B6 mice are often used as the default mouse model, but may not represent the response that is typical for the species, therefore reducing the translatability of results to humans. Adequate in vivo scientific evidence is of critical importance because this type of data is used in the periodical revision of air quality standards, aiming at establishing acceptable air pollution exposure levels for general and susceptible populations. Phenotypic differences between B6 and D2 mice will segregate in the B6-by-D2 recombinant inbred (BXD) family of mice, allowing the identification of the genetic loci responsible for these differences. The BXD family consists of more than 140 recombinant inbred strains created by intercrosses between B6 and D2 strains [63]. The more than 6 million genetic differences between the two parental strains are distributed and recombined among the 140 recombinants. This would in turn provide insight into the mechanisms of action and reasons for differential susceptibility to DE toxicity. These results pave the way for follow-up research to include 30–40 BXD strains and thus, enable gene mapping to nominate candidate genes that underlie individual differences in susceptibility to neuroinflammatory and possible neurodegenerative responses to inhalation exposure to DE. Such study has high translational potential because many mouse and human genes are highly conserved, and the two genomes are more than 90% syntenic.

## 4. Materials and Methods

### 4.1. Animals

The animals for this study were male and female C57BL/6J (B6) and DBA/2J (D2) mice purchased from Jackson Laboratories (Bar Harbor, ME, USA), which were exposed to either filtered air or diluted DE at 62–63 weeks of age. We elected to use aged mice in this study as it was previously demonstrated that elderly people are more susceptible to the adverse outcomes of air pollution, including neurological effects [44]. Indeed, it was showed that the locomotor activity and the gait velocity of 12 months old male C57BL/6 correlates with an elderly human being aged 60 or older [64]. Thus, supporting that the ~15 months old mice we used in our study were aged mice at the time of tissue collection. All mice were housed in an AAALAC-approved animal care facility at the School of Veterinary Medicine of the Louisiana State University under a 12-h light/dark cycle (from 6:00 am to 6:00 pm). The mice had access to water and food ad libitum, except during the 2-h exposure periods. Mice were housed and handled in accordance with the NIH Guide for the Care and Use of Laboratory Animals. All procedures and protocols were approved by the Louisiana State University Institutional Animal Care and Use Committee.

### 4.2. Diesel Exhaust Exposures

Exposures to DE were conducted in 0.75 m^3^ inhalation chambers set for 14 air changes per hour. DE was generated by a 6-horsepower (HP) diesel engine (Carroll Stream Motor Company, Oxford, MI, USA) using standard diesel fuel. The chilled DE was mixed with HEPA-filtered compressed air to generate a targeted total particulate matter (TPM) concentration of 0.95 mg/m^3^ (Figure 6). The mice were exposed to diluted DE for 2 h per day, 5 days a week, for 4 weeks. Under these conditions, the targeted exposure concentration of 0.95 mg/m^3^ yields a 24-h average PM_2.5_ concentration of 0.079 mg/m^3^ or 79 µg/m^3^, a concentration within the range of PM_2.5_ (55 to 150 µg/m^3^) measured in critical areas in the United States [14]. Off-line characterization of the exposure included mass concentration of DE, where the TPM was measured gravimetrically. This was achieved by sampling and collecting the aerosols on glass fiber filters (25 mm hydrophilic glass fiber filter with a 0.7 μm pore size; Millipore Sigma, Cat. #AP4002500, Burlington, MA, USA) that were weighed, before and after sampling, on a Sartorius MC5 microbalance (Sartorius, Goettingen, Germany). In real-time, we measured the surface area concentration, via a nanoparticle dosimeter (Partector, Naneos, Windisch, Switzerland), allowing for the determination of the surface area of particles in a size range from 10 nm to 10 μm, corresponding to human lung-deposited surface areas of 0 to 2500 µm^2^/cm^3^ for particles reaching the tracheobronchial region and of 0 to 20,000 µm^2^/cm^3^ for particles reaching the alveolar region. In addition, we determined the level of carbon monoxide (CO) concentration by an infrared spectrometer (Miran Sapphire, Thermo Fisher Scientific, Franklin, MA, USA), as previously reported in Noël et al. [28]. Concentration of CO in the diesel exhaust condition averaged 9.27 ppm (Table 2). Control mice were exposed to HEPA-filtered air in a chamber similar to the one used for the experimental group, but without DE. Both groups were exposed simultaneously. Exposure characterization data are reported in Table 2.

### 4.3. Brain Tissue Harvest

On the day following the last day of DE exposure, the animals were euthanized. The brain was removed and the medial prefrontal cortex was isolated and immediately placed on dry ice. 

### 4.4. RNA Isolation, cDNA Synthesis, and rtPCR

Expression of the proinflammatory cytokine genes for *Il1β*, *Il6*, and *Tnfa* in brain samples (medial prefrontal cortex), DE vs. control was analyzed by qPCR as described by us previously [65,66]. Total RNA was isolated using standard methods. Briefly, total RNA was isolated using Trizol reagent (Invitrogen, Grand Island, NY, USA) and Phase-lock heavy gel (Eppendorf AG, Hamburg, Germany) and purification by RNeasy mini-spin column (Qiagen, Valencia, CA, USA). Reverse transcription of the total RNA(1lg) to cDNA was achieved by SuperScriptTMII RNase and oligo (dT)12–18primers (Invitrogen) in a 20lL reaction. qPCR analysis of glyceraldehyde-3-phosphate dehydrogenase, tumor necrosis factor-alpha (TNF-a), interleukin 6 (IL6), and interleukin 1beta (IL-1b) was performed in an ABI PRISM 7500 sequence detection system (Applied Biosystems, Carlsbad, CA, USA) in combination with TaqMan chemistry. Primers are listed in Table 3.

Relative quantification of gene expression was performed using the comparative threshold (ΔΔC_T_) method [67]. Changes in mRNA expression levels were calculated after normalization to *Tbp*. The ratios obtained after normalization are expressed as fold change over corresponding controls. The numbers of mice by strain, sex, and treatment are presented in Table 4.

### 4.5. Data Analysis

Transcript abundance for *Il1β*, *Il6*, and *Tnf**α* obtained by qPCR was LOG_2_ transformed and analyzed by analysis of variance for three between-subjects variables (strain, DE, sex) experiment. Main effects and interactions were considered statistically significant at α = 0.05. 

### 4.6. RNA-seq Analysis for Genome-Wide Transcript Abundance

Genome-wide, differentially expressed genes (DE vs. control) were analyzed using the next-generation RNA-sequencing method, RNA-seq. At the end of each treatment period the animals (n = 2 from each strain, sex, treatment) were euthanized, the brain was removed and prepared for isolation of the brain, and dissected to yield the medial prefrontal cortex. The tissues were homogenized for 2 min in a Tissue Lyser II (Qiagen, Hilden, Germany) with a speed frequency of 30 r followed by incubating for 5 min. 140 µL chloroform was added to the homogenate, shaken vigorously for 15 s, and centrifuged for 15 min at 12,000× *g* at 4 °C. 280 µL of the upper aqueous layer was then transferred into a new collection tube containing 500 µL of 100% ethanol. The mixture was loaded into an RNeasy Mini column, once with Buffer RWT and twice with Buffer RPE purification. The final RNA was diluted into 50 µL RNase-free later. RNA purity and integrity were determined by NanoDrop One (Thermo Fisher Scientific, Waltham, MA, USA) and Agilent 2100 Bioanalyzer (Agilent Technologies, Santa Clara, CA, USA). RNA with OD260/280 > 1.8 and RIN > 8.0 for library preparation. 

### 4.7. Library Preparation and Sequencing

1 µg of RNA was used for cDNA library construction at Novogene using a NEBNext^®^ Ultra RNA Library Prep Kit for Illumina^®^ (cat# E7420S, New England Biolabs, Ipswich, MA, USA) according to the manufacturer’s protocol. Briefly, mRNA was enriched using oligo(dT) beads followed by two rounds of purification and fragmented randomly by adding fragmentation buffer. The first strand of cDNA was synthesized using random hexamers primer, after which a custom second-strand synthesis buffer (Illumina, San Diego, CA, USA), dNTPs, RNase H, and DNA polymerase I were added to generate the second strand (ds cDNA). After a series of terminal repairs, poly-adenylation, and sequencing adaptor ligation, the double-stranded cDNA library was completed following size selection and PCR enrichment. The resulting 250–350 bp insert libraries were quantified using a Qubit 2.0 fluorometer (Thermo Fisher Scientific, Waltham, MA, USA) and quantitative PCR. Size distribution was analyzed using an Agilent 2100 Bioanalyzer (Agilent Technologies, Santa Clara, CA, USA). Qualified libraries were then sequenced on an Illumina Novaseq Platform (Illumina, San Diego, CA, USA) using a paired-end 150 run (2 × 150 bases). An average of 40 million raw reads were generated from each library.

### 4.8. Data Quality Control and Filtering

The raw reads underwent the following filter to produce clean data: (1) Remove reads containing adaptors; (2) Remove reads containing N > 10%; and (3) Remove reads in which 50% bases have a Qscore (Quality value) <= 5. 

### 4.9. Alignment to the Reference Genome

*Mus musculus* (mouse) reference genome (GRCm38) and gene model annotation files were downloaded from the Ensembl genome browser (https://useast.ensembl.org/ (accessed on 5 May 2022). Indices of the reference genome were built using STAR v2.5.0a [68] and paired-end clean reads aligned to the reference genome. STAR uses the method of Maximal Mappable Prefix to generate a precise mapping result for junction reads.

### 4.10. Quantification of Gene Expression Level

In RNA-seq experiments, gene expression was estimated by the abundance of transcripts (counts of sequencing) that map to a gene or exon. Read counts were proportional to gene expression level, gene length, and sequencing depth. FeatureCount v0.6.1 [69] was used to count the number of reads mapped to each gene. Transcripts Per Million (TPM) were calculated for each gene based on the length of the gene and reads mapped to that gene. In this normalization, the sum of all TPMs (genes-level) is equal to 1,000,000. 

### 4.11. Differential Expression Analysis

Differential expression analyses between two conditions DE vs. control were performed using the R package DESeq2 v1.22.2 [70]. DESeq2 provides statistical routines for determining differential expression in digital gene expression data using a model based on the negative binomial distribution. The resulting P-values were adjusted using Benjamini and Hochberg’s approach for controlling the False Discovery Rate (FDR) [71]. Genes with an FDR *p* < 0.05 found by DESeq2 were then defined as differentially expressed genes (DEGs). 

### 4.12. Gene Set Enrichment Analysis

Gene set enrichment analysis for Gene Ontology (GO, biological process), Kyoto Encyclopedia of Genes and Genomes (KEGG) pathway, and Mammalian Phenotype Ontology (MPO) were analyzed using WebGestalt (http://www.webgestalt.org (accessed on 5 May 2022) [66,72]. All protein-coding genes in the mouse genome were used as the reference gene set. The statistical significance of enrichment between DEGs and the members of known GO terms, KEGG pathways, and MPO categories were calculated based on the hypergeometric test. The Benjamini and Hochberg correction [71] was used for multiple test corrections. A minimum overlap of 5 genes and an FDR cutoff of 0.05 were the criteria to determine significance.

## Figures and Tables

**Figure 1 ijms-23-12461-f001:**
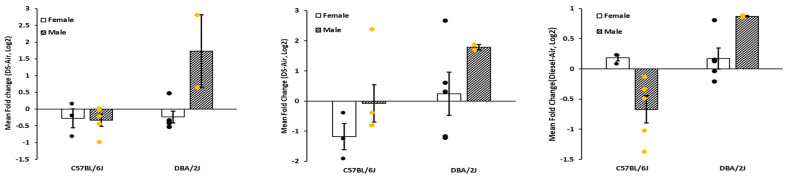
Proinflammatory cytokine gene expression responses to diesel exhaust exposure in prefrontal cortex in male and female C57 and DBA mice. These figures show strain, sex, and treatment Log_2_ mean differences in expression (diesel exposure vs. air) of *Il1β* (**left**) *Il6* (**center**), and *Tnfα* (**right**). Results are expressed as mean ± SEM for expression for *Il1β*, *Il6*, and *Tnfα* by strain, sex, and DE.

**Figure 2 ijms-23-12461-f002:**
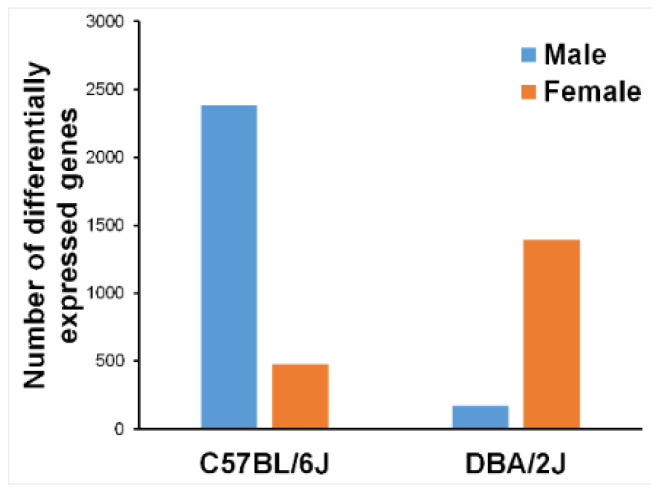
Counts of differentially expressed genes, DE minus air by strain and sex.

**Figure 3 ijms-23-12461-f003:**
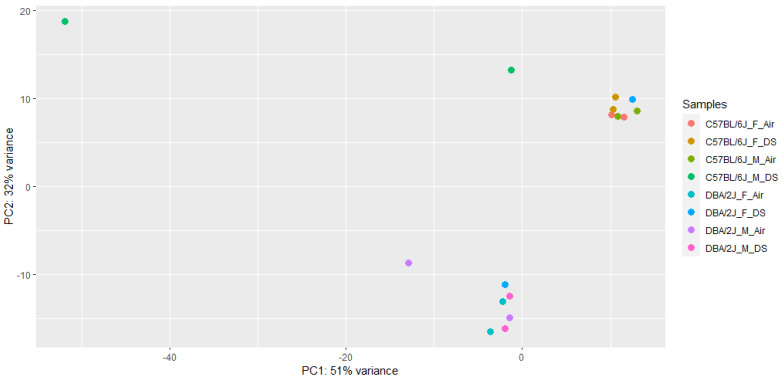
PCA plot of RNA-seq data, with individuals colored by strain (C57BL/6J or DBA/2J), sex (M or F), and treatment (DS or Air) combination. Note the one outlier sample that lies far to the top left of the figure, and so was removed from downstream analyses.

**Figure 4 ijms-23-12461-f004:**
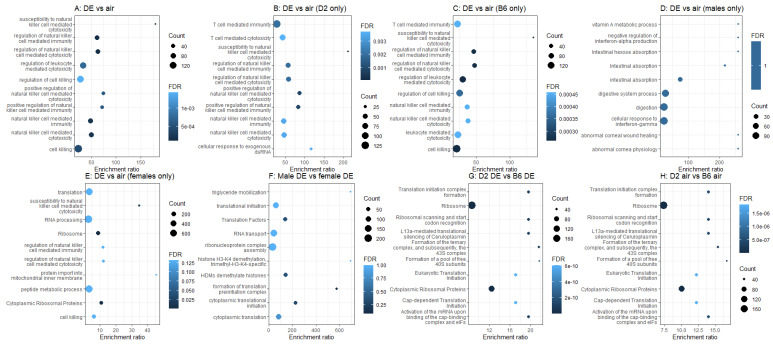
Top 10 most significantly enriched annotations for each pairwise comparison between groups. Comparisons are between diesel exposed (DE) and control (air), DBA/2J (D2) and C57BL/6J (B6), and sex (male vs. female). The x-axis represents the enriched ratio and the y-axis represents enriched pathways/terms. The size of each dot represents the number of genes and the color indicates the FDR. The enriched ratio is defined as the number of observed genes divided by the number of expected genes from the annotation category in the gene list.

**Figure 5 ijms-23-12461-f005:**
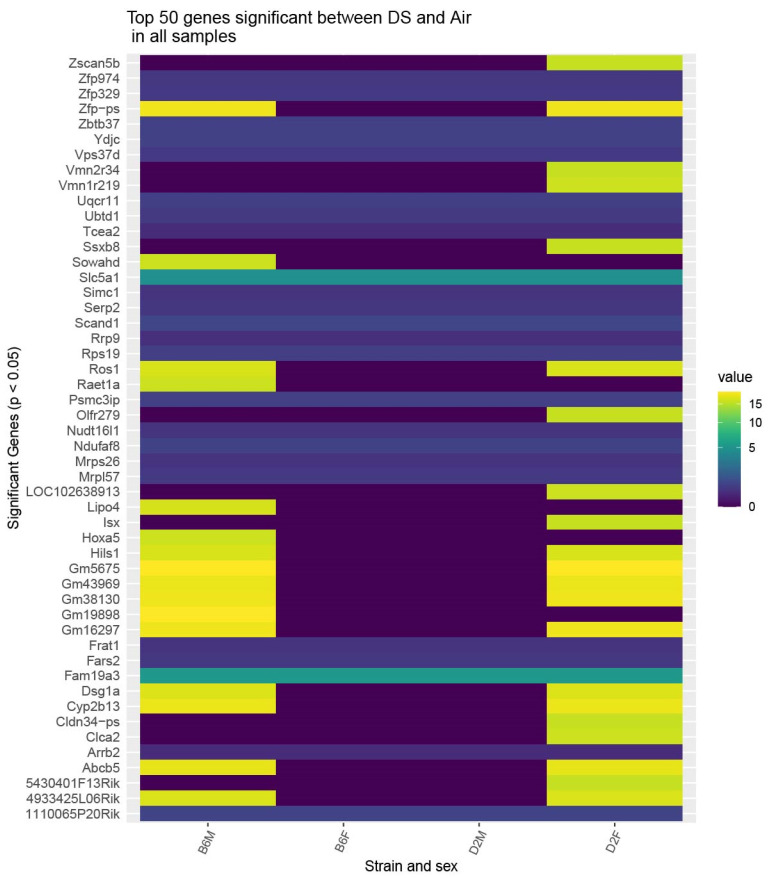
Heatmap of the top 50 genes differentially expressed between all diesel exposed (n = 7) and control (air n = 8) animals, split by strain (C57BL/6J = B6, DBA/2J = D2) and sex (male = M, female = F). Some genes, such as *Slc5a1* and *Fam19a3*, evinced a consistent fold change in all subsets, whereas others are driven by only a subset, often B6M or D2F. Data are absolute values of differential expression, diesel exposure vs. control. All differential expressions were significant at *p* < 0.05 or less.

**Figure 6 ijms-23-12461-f006:**
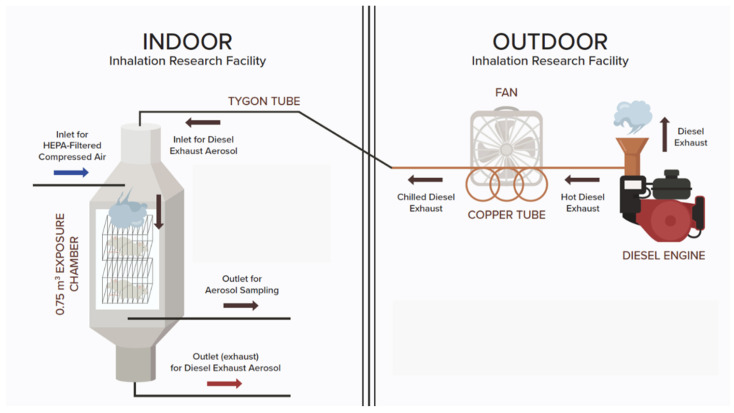
Schematic of the experimental set-up. Diesel exhaust aerosol generation and mouse exposure systems. This schematic is not to scale.

**Table 1 ijms-23-12461-t001:** Proinflammatory cytokine gene expression in animals exposed to air, i.e., the controls for Figure 1.

Strain	Sex	Gene	Mean Fold Change	s.e.m.	n
C57	F	*Il1b*	0.001	5.22 × 10^−5^	3
	M	*Il1b*	0.001	0.00001	5
D2	F	*Il1b*	0.002	0.00002	5
	M	*Il1b*	0.001	4.97 × 10^−5^	2
B6	F	*Il6*	3.76 × 10^−5^	1.34 × 10^−5^	3
	M	*Il6*	1.43 × 10^−5^	2.46 × 10^−6^	5
D2	F	*Il6*	1.47 × 10^−5^	3.13 × 10^−6^	5
	M	*Il6*	2.61 × 10^−6^	1.09 × 10^−6^	2
B6	F	*Tnfa*	0.007	0.00004	3
	M	*Tnfa*	0.006	0.0014	5
D2	F	*Tnfa*	0.008	0.00077	5
	M	*Tnfa*	0.006	0.0006	2

**Table 2 ijms-23-12461-t002:** Diesel exhaust exposure characterization data measured inside the exposure chambers in the breathing zone of the animals.

	Filtered Air	Diesel Exhaust
Temperature (°C ± SEM)	22.5 ± 0.5	22.8 ± 1.0
Humidity (%RH ± SEM)	36.9 ± 3.7	26.4 ± 0.9
CO (PPM ± SEM)	---	9.27 ± 1.21
Average total particulate matter (TPM) concentration measured throughout the daily exposures (mg/m^3^ ± SEM)	---	0.95 ± 0.3
Particle surface area (μm^2^/cm^3^ ± SEM)	---	8.99 ± 3.47

Data are expressed as mean +/− standard error of the mean (SEM).

**Table 3 ijms-23-12461-t003:** Proinflammatory gene expression primers.

Gene	Orientation	Labeled Name	Sequence
**IL1beta**	Downstream	mIL1bpro38Le	AGT TGA CGG ACC CCA AAA G
	Upstream	mIL1bpro38Ri	AGC TGG ATG CTC TCA TCA GG
**IL6**	Downstream	mIL6pro55Le	CGC TAT GAA GTT CCT CTC TGC
	Upstream	mIL6pro55Ri	TTG GGA GTG GTA TCC TCT GTG
**TNFa**	Downstream	mTNFapro25le	CTG TAG CCC ACG TCG TAG C
	Upstream	mTNFapro25ri	TTG AGA TCC ATG CCG TTG
**TBP**	Downstream	mTBPp107le	TCT GGG TTA TCT TCA CAC ACC A
	Upstream	mTBPp107Ri	GGG GAG CTG TGA TGT GAA GT

**Table 4 ijms-23-12461-t004:** Numbers of mice by strain, sex, and treatment for the proinflammatory cytokine gene expression experiment.

	B6		D2	
	Female	Male	Female	Male
Air	3	5	5	2
Diesel	3	5	5	3

## Data Availability

All data are available upon request.

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
