# Peer review of "Genomic Basis for Individual Differences in Susceptibility to the Neurotoxic Effects of Diesel Exhaust"

_ijms, 2022, doi:10.3390/ijms232012461_

Round 1

Reviewer 1 Report

This manuscript tried to investigate the neuroflammatory markers and whole transcriptome information relevant to sub-chronic inhalation exposed DE in two inbred mouse stains and both sexes. The findings are important for further neurotoxicity studies of DE. However, there are several problems should be addressed before considering to be accepted:

1.       The introduction should be segmented to make reader convenient.

2.       The reason of choosing these two mouse stains should be explained.

3.       The number of two stains and sexes are not same. Author should explain.

4.       It is limited to assess only the proinflammatory cytokine gene expressions to set as the neuroflammatory markers.

5.       The discussion also should be segmented.

6. The differentially expressed genes should be validated and significantly enriched pathways should be discussed in more details when no confirmed data.

Author Response

This manuscript tried to investigate the neuroflammatory markers and whole transcriptome information relevant to sub-chronic inhalation exposed DE in two inbred mouse stains and both sexes. The findings are important for further neurotoxicity studies of DE. However, there are several problems should be addressed before considering to be accepted:

We thank the reviewer for her or his comments.  Each is addressed below:

  1. The introduction should be segmented to make reader convenient. Done as requested
  2. The reason of choosing these two mouse stains should be explained.This was presented in the discussion. The two strains are the progenitors of the 140 BXD recombinant inbred strains. Future work will be conducted in 30+ of these strains to allow for identification of phenotypically relevant candidate genes.
  3. The number of two stains and sexes are not same. Author should explain. If the reviewer means unequal n among the cells, this is not a threat to internal validity of the findings. Most of our studies like this are published with unequal n among the cells.
  4. It is limited to assess only the proinflammatory cytokine gene expressions to set as the neuroflammatory markers. These three genes are our tried and true inflammatory markers and have served us well in our Gulf War illness studies.
  5. The discussion also should be segmented. Done as requested.
  6. The differentially expressed genes should be validated and significantly enriched pathways should be discussed in more details when no confirmed data.

We thank the reviewer for their comment but disagree with the necessity of PCR validation of RNA-seq results (in contrast to microarray results, which often required validation). A number of studies, reviewed by Coenye (2021), have shown that the concordance between RNA-seq and qPCR is high, and non-concordant genes tend to be small and low abundance. Given that our enrichment analysis after RNA-seq is based on tens or hundreds of genes, rather than concentrating on a handful of genes, gives us confidence that our results are not being driven by these kinds of small, low abundance genes that might be discordant between RNA-seq and qPCR.  

There is little evidence that qPCR on the same samples as were used for RNA-seq will give different results. True validation would require replicating the results in a different cohort of animals (either by RNA-seq or qPCR), rather than retesting the same samples using a different method.  

Although we do discuss a small number of genes in the discussion section, these are not the main results of our analyses.   

Coenye, T. (2021). Do results obtained with RNA-sequencing require independent verification? Biofilm 3, 100043. doi:10.1016/j.bioflm.2021.100043. 

Reviewer 2 Report

Vehicular air pollution is an environmental toxicant that can have several health consequences, such as decreased respiratory and cardiovascular function and an increased incidence of age-related dementia and neurodegenerative diseases such as Alzheimer's disease. Moreover, emerging evidence suggests that neuroinflammation is involved in both depression and neurodegenerative diseases. The kynurenine pathway, generating metabolites which may play a role in pathogenesis, is one of several competing pathways of tryptophan metabolism. Thus, in numerous experimental models, likewise pollutants, natural antioxidants is shown  to induce hormetic dose responses that are not only common but display endpoints of biomedical and clinical relevance. These hormetic responses are mediated via the activation of nuclear factor erythroid- derived 2 (Nrf2) antioxidant response elements (AREs) and, as such, are characteristically biphasic, well integrated, concentration/dose dependent, and specific with regard to the targeted cell type and the temporal profile of response. In experimental disease models, the polyphenol-induced hormetic activation of Nrf2 was shown to effectively reduce the occurrence and severity of a wide range of human-related pathologies, including Parkinson's disease, Alzheimer's disease, stroke, age-related ocular damage, chemically induced brain damage, renal nephropathy, as well as obesity and diabetes, amongst others, while also enhancing stem cell proliferation. Interestingly, the mechanistic profile of natural antioxidants is similar to that of numerous other hormetic agents, indicating that activation of the Nrf2/ARE pathway is probably a central, integrative, and underlying mechanism of hormesis itself. The Nrf2/ARE pathway provides an explanation for how large numbers of agents that both display hormetic dose responses and activate Nrf2 can function to limit age-related damage, the progression of numerous disease processes, and chemical- and radiation- induced toxicities. These findings extend the generality of the hormetic dose response to include hormetic nutrients  as powerful chemical activators of Nrf2 that are cited in the biomedical literature and therefore have potentially important public health and clinical implication. Thus, interplay and coordination of redox interactions with endogenous and exogenous antioxidant defence systems  is an emerging area of reserach interest in  anti-inflammatory anti-degenerative therapeutics. Moreover, particular attention has been given to providing an assessment of the quantitative features of the dose-response relationships and underlying mechanisms that could account for the biphasic nature of the hormetic response after exposure to redox active agents, such as environmental oxidants involved to have significant impact on  the biology of resilience mechanisms with inflammatory/antinflammatory consequences on tissue stress resistance. The hormetic dose response should be seen as a reliable feature of the dose response for oxygen free radicals and their redox regulated transcriptional factors  as well as  antioxidant compounds and appears to have an important impact  on pathophysiology and stress resistance mechanisms to oxidative and inflammatory insult and degenerative/proliferative mechanisms in the  regulation of responses  citoprotective interventations.

This is an interesting paper.  The study is well-conceived and well-executed. This reviewer is satisfied with the significance of this study, the care in which the study was performed, and the implications of the results for human health.  However, although the results presented are convincing, the work raises some concerns which will need to be addressed. The questions posed are of extremely high interest, but the paper does not give adequate definitive information, therefore pending addressing some major question is possible to accept for publication.

Minor concerns:

1. Preconditioning signal leading to cellular protection through Hormesis (mithormesis) is an important redox dependent aging-associated to free radicals species accumulation, inflammatory responses involved in the pathophysiology of a number of inflammation-driven disease states.  This aspect should be highlighted in the discussion and references properly added (Calabrese et al., 2010, Antiox. Redox Signal 13,1763; Perluigi M., et al. Journal of Neuroscience Research 888, 3498 - 3507, 2010).

2. Given the relationship between vitagene network and its possible biological relevance in resilience mechanisms underlying pathophysiology of  oxidative stress-driven dysmetabolic as well as neurodegenerative diseases, relevant to mitochondrial medicine approaches Authors while interpetrating results can illustrate appropriately this aspect and make proper connection in the discussion. These references can be quoted (Calabrese et al., Nature Neurosci., 2007 8, 766; Mancuso C., et al., (2006) Redox Rep. 11, 207-213; Drake J., et al., (2003) Journal of Neuroscience Research 74, 6, 917 - 927).

Author Response

(The authors gave the same response as above.)
